# Technical Note: Resolution Enhancement of Flood Inundation Grids

Seth Bryant[1,2], Guy Schumann[3], Heiko Apel[1], Heidi Kreibich[1], and Bruno Merz[1,2]

[1]GFZ German Research Centre for Geosciences, Section 4.4. Hydrology, Potsdam, Germany
[2]Institute of Environmental Science and Geography, University of Potsdam, Potsdam, Germany
[3]School of Geographical Sciences, University of Bristol, Bristol, TN, UK

**Correspondence:** Seth Bryant (seth.bryant@gfz-potsdam.de)

**Abstract.**

High-resolution flood maps are needed for more effective flood risk assessment and management. Producing these directly with hydrodynamic models is slow and computationally prohibitive at large scales. Here we demonstrate a new algorithm for post-processing low-resolution inundation layers by using high-resolution terrain models to disaggregate or downscale. The new algorithm is roughly eight times faster than state-of-the-art algorithms and shows a slight improvement in accuracy when evaluated against observations of a recent flood using standard performance metrics. Qualitatively, the algorithm generates more physically coherent flood maps in some hydraulically challenging regions compared to the state-of-the-art. The algorithm developed here is open source and can be applied in conjunction with a low-resolution hydrodynamic model and a high-resolution DEM to rapidly produce high-resolution inundation maps. For example, in our case study with a river reach of $20\ km$, the proposed algorithm generated a $4\ m$ resolution inundation map from $32\ m$ hydrodynamic model outputs in $33\ seconds$, compared to a $4\ m$ hydrodynamic model runtime of $34\ minutes$. This 60-fold improvement in runtime is associated with a $25\%$ increase in RMSE when compared against the $4\ m$ hydrodynamic model results and observations of a recent flood. Substituting downscaling into flood risk model chains for high-resolution modelling has the potential to drastically improve the efficiency of inundation map production and increase the lead time of impact-based forecasts, helping more at-risk communities prepare for and mitigate flood damages.

## 1 Introduction

Over the past decade, there has been significant progress in the development and implementation of flood models for large continental river basins and at global scale. This is due to several factors including the rise in flood-related disaster damages, advancements in computing, and the availability and quality of global datasets (Ward et al., 2020; Nones and Caviedes-Voullième, 2020). As these models underpin risk management activities, ranging from early warning to land-use planning to disaster response, their accuracy and efficiency are important considerations for improving disaster resilience (De Moel et al., 2009).

Because of the computational demands of hydrodynamic models, resolution has been extensively studied and found to be one of the parameters of most importance for accuracy (Horritt and Bates, 2001; Fewtrell et al., 2008; Savage et al., 2016; Papaioannou et al., 2016; Alipour et al., 2022) with most finding inundation area and flood depth overestimated at coarser

resolutions (Saksena and Merwade, 2015; Mohanty et al., 2020; Ghimire and Sharma, 2021; Muthusamy et al., 2021; Banks et al., 2015). In a study comparing fine and coarse models with resolution ranging from $1\ m$ to $50\ m$ and identical roughness, Muthusamy et al. (2021) used separate resolutions for the channel and floodplain. They found an overestimate in water depths and attributed it to the poorly defined coarse river channel (e.g., thalweg depth underestimated or steep bank misrepresentation) and a subsequent reduction in conveyance (Muthusamy et al., 2021).

There are three primary hazard grids included in most flood risk models: Water Depth ($WSH$), Water Surface Elevation ($WSE$), and the Ground Elevations ($DEM$) which can be related by $WSE = DEM + WSH$. Often, $WSE$ or $WSH$ grids are produced from a hydraulic analysis or some model structured on the $DEM$, leading to a natural pairing of resolution, datum, and domain (i.e., the real-world region associated with the model). For large scale studies, the resolution ($s$) of the $DEM$ is relatively coarse (30-100 $m$), resulting from the process and data used to construct the terrain model, or from some post-process upscaling introduced to obtain the resolution desired by the hydraulic analysis (i.e., coarsening model resolution to reduce complexity and runtime). For this latter case, or any case where supplementary fine resolution ($s1$) DEM grids are available, applications like flood damage modelling or impact-based forecasting may benefit from enhancing $WSE$ grids through downscaling or disaggregation to obtain a finer resolution without the need for expensive or unstable hydrodynamic modelling. Unlike super-resolution techniques, which seek a high-resolution image from a single low-resolution image (Dong et al., 2015), flood hazard grid downscaling is a well-posed problem that uses the high-resolution $DEM_{s1}$ and simple hydraulic assumptions to seek the $WSE_{s1}$ grid that would be generated by an otherwise equivalent high-resolution hydrodynamic model.

While many flood risk model studies maintain a single resolution throughout the analysis (Hall et al., 2005; Sairam et al., 2021), examples of both upscaling and downscaling hazard grids are common. Upscaling, where hazard model output grids are post-processed to coarsen resolution, is generally undertaken to facilitate intersection with some exposure data, which is generally the most coarse data grid in flood risk model chains. This upscaling is achieved either through simple averaging (Seifert et al., 2010; Sieg and Thieken, 2022) or some unspecified method (Thieken et al., 2016; Jongman et al., 2012). Examples of downscaling in the literature may employ it to reverse some earlier coarsening which was applied to improve hydrodynamic model stability or efficiency (Schumann et al., 2014; Sampson et al., 2015) or enhance some remote-sensing derived inundation product (Fluet-Chouinard et al., 2015; Aires et al., 2017).

In the first and only study (we are aware of) to investigate downscaling 2D calibrated hydrodynamic models, Schumann et al. (2014) developed a method using a nearest-neighbour search with $N_4(P)$ adjacency (querying values from only the N, S, E, and W adjacent or neighbouring cells, rather than $N_8(P)$ which queries all eight neighbouring cells) and a search radius of half the coarse resolution. The researchers tested their algorithm using two models: a fine model with a 30 $m$ resolution and a coarse model with a 600 $m$ resolution, each calibrated separately. When comparing the downscaled grid to the results of the fine hydrodynamic model, they discovered an overestimate in water levels and negligible differences in volume. This method substantially improved computation times (compared to hydrodynamic modelling) and provided the basis for some large-scale flood models (Sampson et al., 2015; Bates et al., 2021). The CaMa-Flood project (Yamazaki et al., 2011) has developed a fortran script with a similar algorithm to downscale results of their global river model; however, this script has not been described in any publication we are aware of. With the objective of operationalizing a 2D hydrodynamic model,

Fraehr et al. (2023) developed a modelling framework that integrates a Guassian Process learning model with a low-fidelity hydrodynamic model to yield high-resolution depth and inundation estimates.

As part of their work to enhance the VIIRS (Visible Infrared Imaging Radiometer Suite) $375\ m$ resolution near real-time global flood inundation product, Li et al. (2022) developed a seven-stage downscaling and correction pipeline. Leveraging global datasets for tree cover, land cover, permanent water bodies, and river networks, VIIRS water fractions were first converted to water levels then corrected using simple hydraulic assumptions. These $375\ m$ resolution water level grids were downscaled and converted to depths by intersecting with a global $30\ m$ DEM using a two stage algorithm. In the first stage, a nearest-neighbour search is employed with $N_4(P)$ adjacency and a search radius of one fine pixel starting with the lowest elevation pixel. A similar process is applied to the dry cells in the second stage. This work demonstrates a useful application of downscaling to generate finer resolution flood-related earth observation data; however, the method is not directly applicable to enhancing coarse resolution water grids produced through hydrodynamic modelling because it relies on coarse global data.

While downscaling flood grids is used by many global hazard models, to our knowledge only one study has addressed 2D downscaling of hydrodynamic model results (Schumann et al., 2014) and no studies have provided a methods comparison. Addressing this, our objectives are two-fold: 1) present our newly developed downscaling approach; and 2) evaluate and compare our new approach to the state-of-the-art downscaling approach of Schumann et al. (2014) and two simple algorithms using a data-rich case study.

## 2   Resample Case Framework

To better communicate and understand the challenges and solutions to rescaling flood hazard grids, we adapt the *Resample Case Framework* from Bryant et al. (2023) to classify each cell in the $s2$ domain into one of four cases with similar disaggregation behaviour. Each case is defined by comparing the local coarse water depth value ($WSH_{s2,j}$) to the corresponding fine values ($WSH_{s1,i}$) where cell $j$ is composed of a block of $i$ cells as shown graphically in Fig. 1 and defined explicitly as:

$$case_j = \begin{cases} \text{dry-dry (DD)} & \text{if } max(WSH_{s1,i}) = 0 \\ \text{dry-partial (DP)} & \text{if } WSH_{s2,j} = 0 \text{ and } max(WSH_{s1,i}) > 0 \\ \text{wet-partial (WP)} & \text{if } WSH_{s2,j} > 0 \text{ and } max(WSH_{s1,i}) > 0 \\ \text{wet-wet (WW)} & \text{if } min(WSH_{s1,i}) > 0 \end{cases} \tag{1}$$

where the first part of the $case_j$ label code is determined by the coarse cell ($WSH_{s2}$), and the second letter by the extremes of the fine cells ($WSH_{s1}$). The quadrants in Fig. 1 Panel (b) provide a simple example of four such groups whose corresponding case labels are shown on Panel (a). Because domain resample case classification is dependent on both input and output grids, classification is not directly used in any downscaling algorithms – instead, we use the framework to communicate the process and challenges of downscaling flood hazard grids.

Beginning with the simplest case, dry-dry ($DD$) zones are trivial and can be ignored for flood hazard rescaling operations as they remain unaltered: dry before and after rescaling (i.e., $WSH = 0$). Wet-wet ($WW$) zones are also relatively simple as

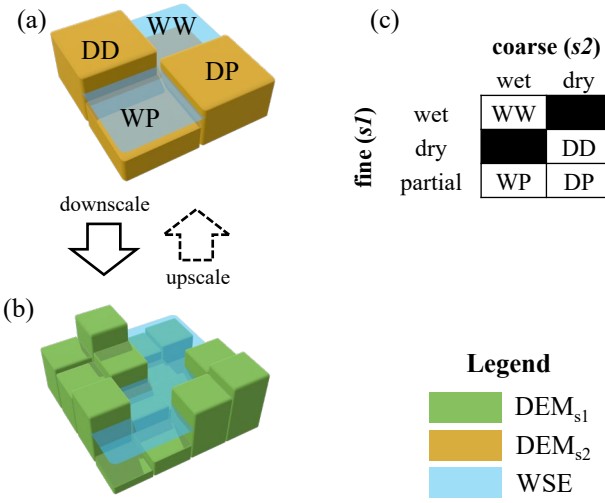

**Figure 1.** Framework for classification of flood hazard resample case. Panel (a) shows conceptual coarse grids and the corresponding resample case calculated from Eq. 1. Panel (b) shows the corresponding fine grids while Panel (c) shows the case label acronyms. D, W, and P stand for *dry*, *wet*, and *partial*, respectively.

the group of $WSE_i$ cells should be roughly equivalent to their parent $WSE_j$ cell; an easy task for classic grid resampling tools like bilinear resampling. Wet-partial ($WP$) zones can be similarly obtained, but with the additional step of removing dry $s1$ cells with an exceedance mask (i.e., cells where $WSE_{s1} < DEM_{s1}$ are set to null). Most difficult is the treatment of dry-partial ($DP$) zones which require some propagation or searching beyond the original coarse ($s2$) inundation footprint. In fact, this propagation challenge is a similar problem to that of a classic 2D hydrodynamic model; however, downscaling must

employ simplifying assumptions to maintain advantageous computation times. This requires more sophisticated algorithms to propagate wet cells laterally like horizontal projection. Such horizontal projection may introduce artifacts like isolated flooding where disconnected groups of cells in low-lying terrain are shown as flooded (e.g., behind levees). Because high-resolution hydrodynamic models generally employ cell-connected routines, this isolated flooding can be considered erroneous within the paradigm of hydrodynamic downscaling. However, this paradigm can be a poor representation of actual flood behaviour in

areas with high groundwater connectivity or imperfect flood defences. In summary, the downscaling problem can be broken into three zones: the first two ($WW$ and $WP$) are relatively simple and, while some alternate approaches are possible, we do not expect large differences in performance in these zones. The final dry-partial ($DP$) zone is more challenging and we therefore expect differences in treatment and performance between algorithms in this zone.

## 3    Methods

To validate and compare the novel resolution enhancement or downscaling algorithm, we first compute the requisite coarse resolution ($s2 = 32m$) input grid ($WSE_{s2}$) using a calibrated hydrodynamic model. Using this input grid ($WSE_{s2}$) and a fine-resolution terrain layer ($DEM_{s1}$) we apply the novel downscaling algorithm to compute a fine-resolution enhanced grid ($WSE_{s1=4m}$). Using the same inputs, we then compute similar enhanced grids for the state-of-the-art downscaling algorithm from Schumann et al. (2014) and two simple algorithms representing solutions of minimum complexity. These enhanced grids

($WSE_{s1}$), along with results from a calibrated fine resolution hydrodynamic model ($s1 = 4m$), are then evaluated against the maximum inundation extents and high water marks observed during a 2021 flood in Germany to validate and demonstrate the performance improvements of the novel algorithm.

### 3.1    Novel CostGrow Algorithm

The novel *CostGrow* algorithm employs the four phases summarized in Fig. 2: 1) grid resampling; 2) least-cost mapping;

3) filtering high-and-dry sub-grid cells; and finally 4) an isolated-cell filter; all of which are parameterless. In the first grid resampling phase, various techniques have been developed by others for applications in image analysis and spatial analysis (Bierkens et al., 2000) with bilinear being the most common for terrain manipulations in hydraulic applications (Heritage et al., 2009; Muthusamy et al., 2021) as it provides a smooth result while preserving centroid values. For downscaling, bilinear resampling computes the $s1$ value from the four adjacent $s2$ centroid values weighted by distance as seen in Fig. 2a (notice

the $s2$ values are preserved by the center $s1$ cell). *CostGrow* implements bilinear resampling from the popular spatial analysis package GDAL (GDAL/OGR contributors, 2022). In the second phase, the resampled grid is extrapolated using a cost-distance analysis, a common GIS algorithm for computing the path of least cost, determined by weighting distance and some cost map to obtain the *effective distance* from source cells to sink cells Foltête et al. (2008). For this study *CostGrow* implements a cost-distance routine with $N_8(P)$ adjacency and a neutral cost surface (Lindsay, 2014, CostAllocation). This first maps the

dry portion of the domain in terms of catchment areas for each boundary cell $WSE_{s1,i}$ from the previous phase, then maps the corresponding boundary $WSE_{s1,i}$ cell value to each of its catchments. In effect, this grows or horizontally projects each $WSE_{s1,i}$ boundary cell value outwards, filling the dry domain with the $WSE_{s1,i}$ values that are closest in distance. For the toy example shown in Fig. 2b, this is a simple extrapolation onto the dry right-side of the domain. Future implementations could employ a non-neutral cost surface to incorporate levees or some other flood obstructions into the analysis. In the third phase,

high-and-dry cells are filtered from this cost-distance map by comparing cell-by-cell to the terrain values (where $DEM_{s1} > WSE_{s1,i}$ set $WSE_{s1,i} = NULL$) as shown by the blank cells in Fig. 2c. This often results in many isolated pockets of flooding in low-lying areas shown beyond the initial contiguous $WSE_{s2}$ flood (see Fig. 2c red circle). In the final phase, these isolated or disconnected groups of flooded cells are filtered from the result such that only the largest or main flooded water body remains. To accomplish this, the filtered grid is converted to a binary inundation grid, from which each contiguous clump

is identified and ranked according to size (Lindsay, 2014, Clump) (see Fig. 2d). From the largest clump, an inverted mask is generated and applied to the water level grid to remove isolated flooding cells from the result.

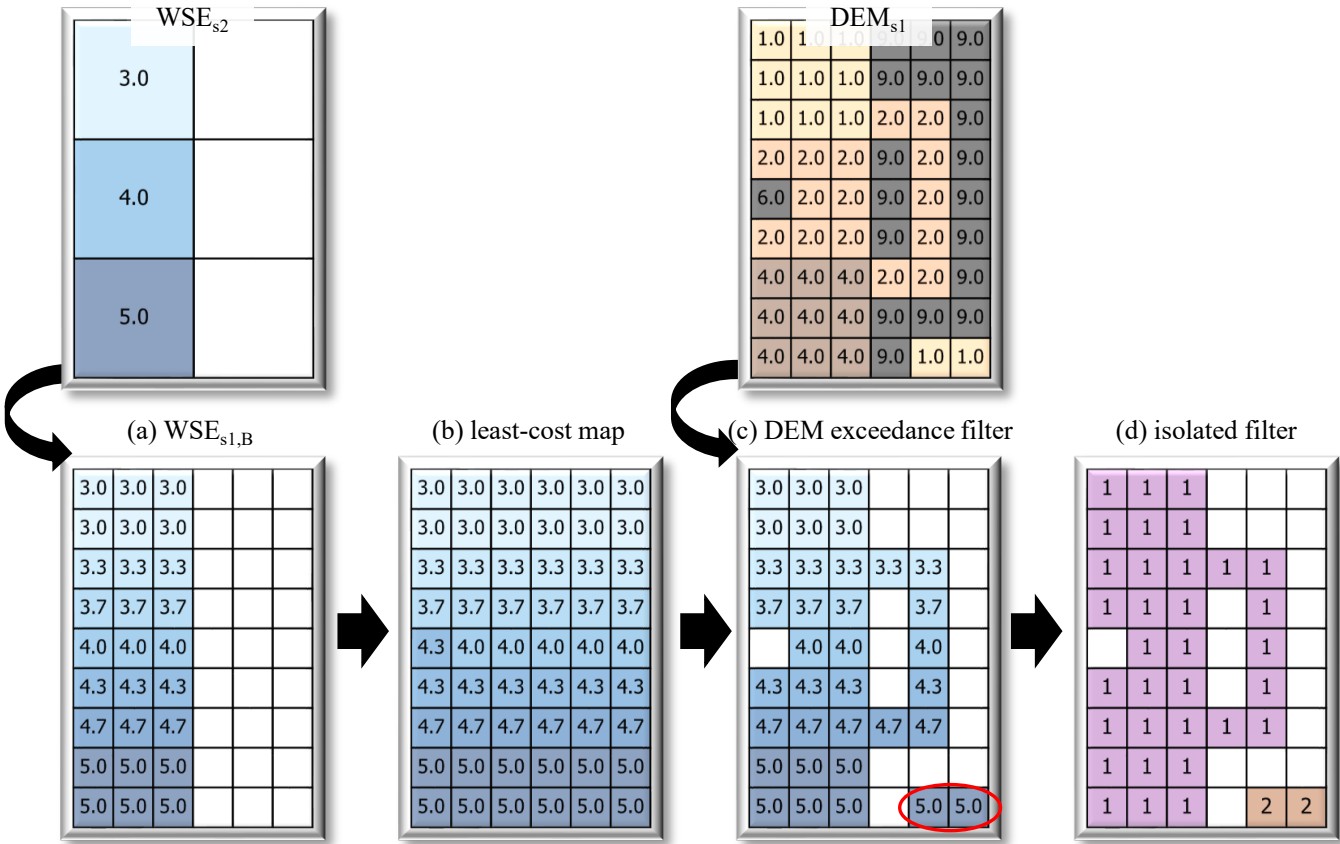

**Figure 2.** Toy example of the novel *CostGrow* downscaling algorithm showing $WSE$, $DEM$, and clump analysis grids for the four phases: a) bilinear resampling of coarse water level $WSE_{s2}$ grid; b) extrapolation using least-cost mapping from $WSE_{s1,B}$ values; c) application of DEM exceedance mask to filter high-and-dry cells; and d) results of clump analysis, from which the isolated mask is generated to filter all but the largest group (1 in this case). See text for details.

## 3.2 Validation and Comparison

### 3.2.1 Case Study, Data, and Hydrodynamic Modelling

To evaluate the aforementioned downscaling algorithms, data obtained from the July 2021 flooding of the Ahr River in Germany
is used. This was the most extreme flood event to hit the region in living memory, with precipitation exceeding a 500-year return period (Dietze et al., 2022), a difficult to estimate peak discharge (Vorogushyn et al., 2022), and 134 casualties in the Ahr valley (Szönyi M. and Roezer V., 2022). The data used for this study is summarized in Table 1 and Fig. 3.

To construct the coarse water grid for use as an input in downscaling ($WSE_{s2}$) and a second grid for validation and comparison ($WSE_{s1}$), coarse ($s2 = 32m$) and fine ($s1 = 4m$) resolution twin hydrodynamic models are calibrated to the observed
inundation extents using the Critical Success Index. The hydrodynamic models are constructed in the 2D raster-based *RIM2D*

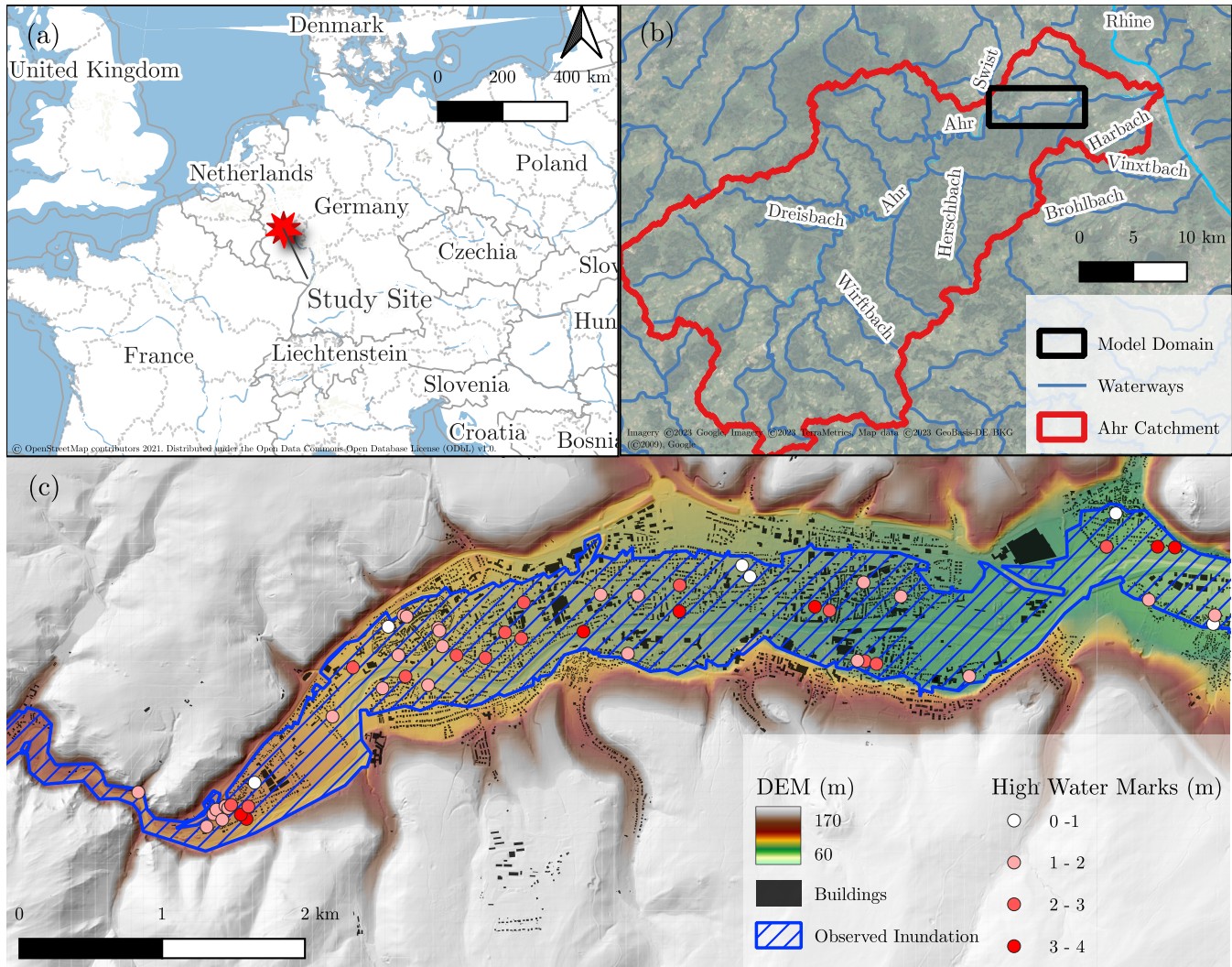

**Figure 3.** Study site maps showing: (a) location map; (b) Ahr catchment map; and (c) downscaling domain with main datasets (see Table 1 for descriptions).

framework (Apel, 2023) and run on a Tesla P100 GPU. RIM2D implements a simplified version of the shallow water equations after Bates et al. (2010). A reconstructed hydrograph is used for the upstream boundary condition and other model parameters are described in Apel et al. (2022). The model terrain is generated through bilinear resampling of the bare earth DEM (Table 1) to the target resolution. For the treatment of urban areas, blocking-out buildings has been shown to be a more accurate way to represent buildings within a relatively high-resolution hydrodynamic model (Bellos and Tsakiris, 2015). In coarse models however, especially where the building size (and space between) is smaller than a grid-cell, blocking-out reduces model performance, especially in and around buildings. Regardless, blocking-out could be included into a downscaling algorithm (and the

**Table 1.** Summary of data used in case study

| type | metadata | ref. |
|---|---|---|
| DEM | 0.5 m resolution bare earth DEM created from aerial LiDAR survey from September 22 to October 24, 2021 in twelve sessions with a RIEGL scanner LMS-VQ780i with 20 points/$m^2$ achieved. | (Milan Geoservice GmbH, 2023) |
| High water marks | 75 high water marks at buildings reported by residents. | (Apel et al., 2022) |
| Inflow hydrograph | 30 hour hydrograph at Altenahr gauge with maximum depth of 10.2 m reconstructed by Environmental Office of the federal state Rhineland-Palatinate. | (Apel et al., 2022) |
| Building locations | Building footprint polygons downloaded from OSM on 2022-11-14. | (OpenStreetMap contributors, 2022) |
| Observed inundation | Polygon of maximum flood extents compiled from an aerial survey on July 16th and 20th and a second survey on July 24th and 29th. | (Landesamt für Umwelt Rheinland-Pfalz, 2022) |
| Land cover | Gridded land cover inventory reflecting 2017-2018 conditions and updated in 2020. | (Copernicus Land Monitoring Service, 2018) |

high-resolution validation model); however, as this requires a more complicated algorithm and is not included in the current state-of-the-art methods against which we compare, we opted to avoid blocking-out and instead apply a separate roughness coefficient to built-up areas in both models (and all downscaling algorithms) to capture the blocking effects of buildings (a.k.a. the "urban porosity approach").

To obtain accurate maximum $WSE$ grids from the twin hydrodynamic models, a calibration routine is used to optimize model roughness using the Critical Success Index (CSI) (see Table S2 for definition) of the maximum simulated inundation calculated against the observed inundation from Table 1. Two unique Manning's roughness values (built-up and channel/flood-plain) are treated as free parameters for each model and optimized while a third roughness value for forested areas is held fixed ($n = 0.2 \frac{s}{m^{1/3}}$) as Apel et al. (2022) showed this third region to have negligible influence. The three roughness values are spatially allocated according to land cover (Table 1) as described in Apel et al. (2022). Finally, the optimal roughness values are obtained using a mix of trial-and-error and the Newton-Conjugate Gradient algorithm (Nocedal and Wright, 2006; Virtanen et al., 2020) to optimize the Critical Success Index (CSI).

The best performing effective roughness values for the twin hydrodynamic models $s2 = 32m$ and $s1 = 4m$ is 0.867 and 0.175 $\frac{s}{m^{1/3}}$ for urban areas and 0.089 and 0.133 $\frac{s}{m^{1/3}}$ for channel areas with a CSI of 0.885 and 0.914 respectively as shown on Fig. S1 and S2. These counterintuitive relative roughnesses are a result of differences in floodplain-channel dynamics necessary to match the observed inundation footprint between the two models. In the coarse hydrodynamic model, the river channel is poorly represented by the 32m resolution which is roughly 3 times larger than the channel. Thus, the flow in the

channel, as well as the channel-floodplain interactions, show different dynamics compared to the more realistic fine resolution model. The calibration routine compensates for these differences with the disproportionate roughness values reported above. However, as our focus is on downscaling performance, the less-accurate representation of channel dynamics and water levels (as opposed to inundation extents) provided by the coarse model are inconsequential considering we apply the fine and coarse hydrodynamic model results comparatively in all scenarios. Additional figures, performance measures, and discussion for the calibration are provided in the supplement. To remove any boundary effects, the hydrodynamic model results are cropped to a smaller domain for the downscaling analysis (13.4 x 6.6 km to 8.9 x 3.5 km; see Fig. S3).

### 3.2.2 Downscaling Algorithms for Comparison

To demonstrate the performance of the novel *CostGrow* algorithm relative to similar algorithms, two simple algorithms representing solutions of minimum complexity and the state-of-the-art from Schumann et al. (2014) are described below and included in the comparison.

The first simple algorithm considered here is a bilinear grid resampling (see *Resample* in Fig. 4c) which is identical to the above described first phase of the novel *CostGrow* algorithm. This algorithm is the only one considered that does not make use of the fine resolution terrain values ($DEM_{s1}$) and therefore carries an obvious limitation in wet-partial ($WP$) regions, where sub-grid high-and-dry ground elevations may be present within a wet coarse cell (see red circle in Fig. 4a). To address this, the second simple algorithm we consider (see *TerrainFilter* in Fig. 4d) builds and applies a terrain exceedance mask ($WSE_{s2} < DEM_{s2}$) which removes those cells where depths are negative from the resulting $WSE_{s1}$. Neither of these simple algorithms treat cells outside the wet coarse domain (i.e., the dry-partial ($DP$) zone remains dry).

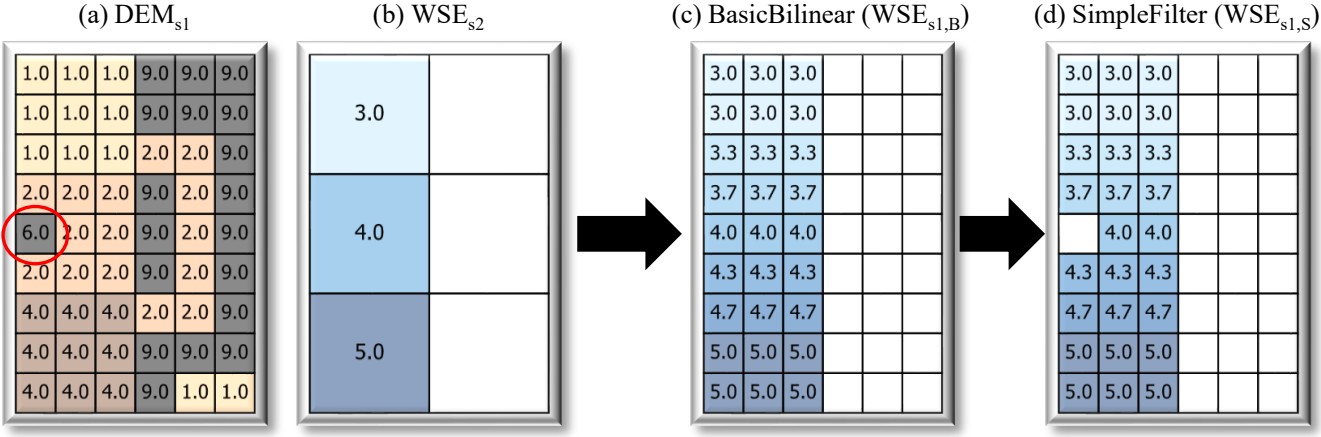

**Figure 4.** Toy example of simple flood downscaling inputs and algorithms showing: a) input fine resolution terrain grid; b) input coarse resolution water level grid; c) *Resample* downscaling result; and d) *TerrainFilter* downscaling result described in the text.

To the best of our knowledge, Schumann et al. (2014)'s method (*Schumann14*) is the state-of-the-art in 2D flood grid downscaling algorithms. This algorithm was developed to downscale 1D/2D hybrid inundation model results from $600\ m$ to

30 $m$ by employing a two tier approach: first, the 1D channel regions are downscaled assuming a water surface plane between

sections; second, 2D floodplain regions are downscaled using a nearest-neighbour search. For our study, we focus on the

floodplain portion of the algorithm for which the source code was provided to the study team and which has roughly the three

steps shown in Fig. 5. First, a search zone is built using a buffer of width one half the coarse resolution around all wet cells

in the coarse domain (i.e., wet-partial (WP) and wet-wet (WW) regions). An alternate buffer distance parameter is possible,

but here we select the same parameter value as Schumann et al. (2014). Second, within the search zone, each fine ($s1$) cell

searches for the nearest coarse ($s2$) cell using a nearest neighbour *city-block* (also called *Manhattan*) search algorithm which

replicates the same $N_4(P)$ adjacency used by their inundation model. Finally, the $WSE_{s1}$ nearest-neighbour search result is

combined with a simple grid resample (also using nearest-neighbour) and a terrain exceedance mask is applied. While this

algorithm improves upon the simple approaches, the blocky $WSE_{s2}$ values remain in the fine $WSE_{s1}$ result, isolated flooding

artifacts are introduced (red circle in Fig. 5d), and the per-cell nearest-neighbour search is computationally expensive. Finally,

the algorithm was originally written in the *MATLAB* programming language and was not made public or widely shared.

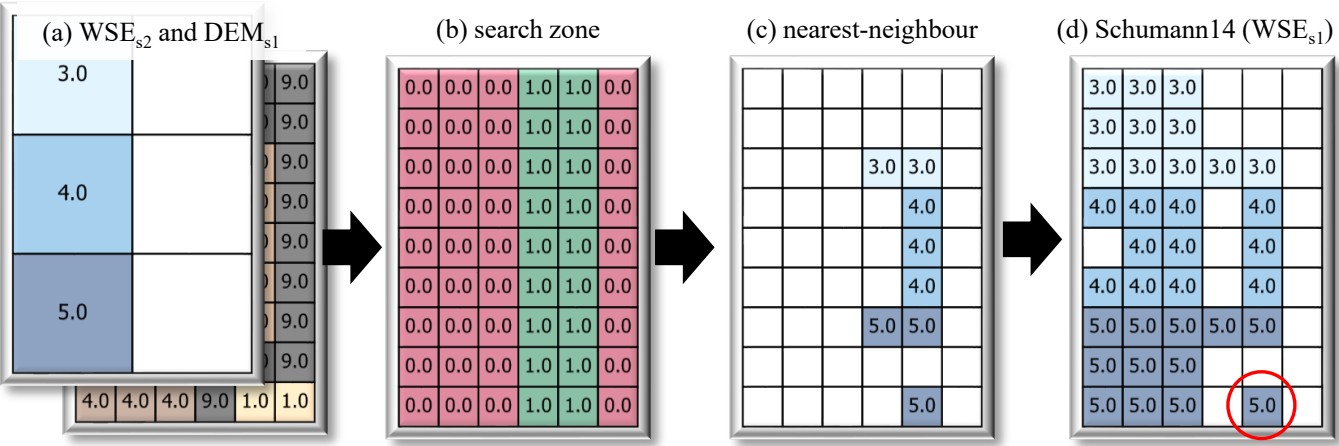

**Figure 5.** Toy example of Schumann et al. (2014)'s floodplain downscaling algorithm showing: a) input fine resolution terrain grid and coarse resolution water level grid (see Fig. 4a); b) search zone; c) nearest-neighbour search result; and d) $WSE_{s1}$ downscaling result. See text for details.

Compared to the above *Schumann14* algorithm, we expect three main advantages from the novel *CostGrow*. First, *CostGrow* should be substantially more computationally efficient by avoiding the cell-by-cell nearest neighbour search. Second, *Cost-Grow* should be slightly more accurate in reproducing fine resolution inundation extents by avoiding the fixed search radius

and including an isolated flooding filter. Third, the accuracy of water levels should improve due to the incorporation of the aforementioned inundation extent mechanisms in dry-partial regions and the replacement of the initial nearest-neighbour re-sampling with bilinear resampling. These expectations are tested below using a case study. Table 2 provides a brief summary of all the downscaling algorithms considered by this study and the two hydrodynamic models used.

**Table 2.** Downscaling algorithms and hydrodynamic models included in this evaluation showing resample $case_j$ applicability (see Fig. 1) and case study runtime to achieve output grids ($WSE_{s1=4m}$ for all except *Hydro. (s2)* which outputs $WSE_{s2=32m}$).

| name | $case_j$ | runtime (secs) | method desc. |
|------|----------|----------------|--------------|
| Hydro. (s1) | n/a | 1917.0 | fine ($s1 = 4m$) hydrodynamic model |
| Hydro. (s2) | n/a | 32.0 | coarse ($s2 = 32m$) hydrodynamic model |
| CostGrow | WW, WP, DP | 1.0 | *TerrainFilter* plus cost-distance mapping and isolated-cell filtering |
| Resample | WW | 0.1 | simple bilinear grid resampling |
| TerrainFilter | WW,WP | 0.2 | *Resample* plus filtering of high-and-dry cells |
| Schumann14 | WW, WP, DP | 8.7 | nearest-neighbour search and grid resampling from Schumann et al. (2014) |

## 4   Analysis and Results

Water level grids obtained from the twin calibrated hydrodynamic models and the four downscaling algorithms and their corresponding inundation performance are shown in Fig. 6. For the models, Fig. 6a4 and b4 show the fine ($s1$) parameterization while the results for the *Resample* downscaling algorithm (Fig. 6a0 and b0) show the performance of the coarse ($s2$) parameterization (because this algorithm does not alter the coarse extents). This suggests both parametrizations reproduce the observed inundation well, with the fine ($s1$) obtaining a slightly better CSI as expected considering the complex topography. The coarse ($s2$) model converged on lower water levels and extents to obtain the optimal CSI, as shown by the lower Error Bias (0.205 vs. 1.029) and the calibration contour plots (Fig. S1 and S2).

Fig. 6 suggests there are some discrepancies between the observed inundation, which has a single contiguous inundation extents without holes, and the DEM which contains some micro-topography that likely would have remained dry during the flood (see Fig. 8 point A), suggesting the observed inundation is slightly conservative (i.e., over-estimates the true flood extents). These discrepancies may be attributable to the methods used by Landesamt für Umwelt Rheinland-Pfalz (2022) to map the extents or the earthworks undertaken between the inundation mapping surveys (late July) and the LiDAR survey (October). Regardless, these discrepancies are relatively minor and we consider them negligible for our research objective of evaluating the *CostGrow* algorithm.

Comparing the inundation performance of the downscaling algorithms in general, the more complex algorithms performed better, with the novel *CostGrow* and *Schumann14* algorithms performing similarly. Specifically, the four common inundation performance metrics in Fig. 6 panels b0-b3 show that *CostGrow* and *Schumann14* have nearly identical performance, with all metrics (with one exception) out performing the simple algorithms *Resample* and *TerrainFilter*. The exception being that *TerrainFilter* has the lowest False Alarm rate (0.014), an artifact which can be explained by two concurrent hypothesis: 1) the *TerrainFilter* algorithm does not address dry-partial ($DP$) regions but does filter wet-partial ($WP$) regions, giving it the smallest inundation area and making it the least likely to over-estimate inundation; and 2) the over-estimation bias in the observed inundation discussed earlier favours methods that under-estimate.

The inundation metrics reported in Fig. 6 are sensitive to both the hydraulic character of the study region and the particular domain selected for analysis e.g., we expect a broad-flat floodplain or different boundary conditions to yield different metric values. For the results reported here, we selected the domain (8.9 x 3.5 km) by balancing hydraulic continuity, controlling for boundary effects, and computational cost; however, other similar domains were tested during study preparation and the relative ranking of performance between the downscaling algorithms was found to be consistent, with *CostGrow* outperforming *Schumann14* slightly for some domains. For example, the CSI of the detail area (Fig. 6 blue box) is 0.813 for *CostGrow* and 0.811 for *Schumann14*. This aligns with our expectation that *CostGrow*'s inundation results would slightly outperform that of *Schumann14* given the absence of a fixed search radius and use of an isolated filter in the *CostGrow* algorithm. Regardless, this evaluation suggests that, while gains in computation performance are substantial, gains in standard qualitative inundation performance over *Schumann14* are negligible. Qualitatively however, *CostGrow* generates more physically coherent depth grids in some fringe areas as shown in Fig. 8 and S4.

Examining the water level performance of the hydrodynamic models, Fig. 7 shows that the coarse ($s2 = 32m$; panel a0) and fine ($s1 = 4m$; panel c1) reproduce the observations well. This is remarkable considering the models were calibrated on inundation extents (using CSI), not water levels, and that the water levels are reported by residents. Similar to inundation performance, Fig. 7 also shows the fine model ($s1 = 4m$) performs best while the coarse ($s2 = 32m$) slightly underestimates (see Fig. S3 for a map of $WSE$ differences between $s2 = 32m$ and $s1 = 4m$). Fig. 7 also shows that, like for inundation performance, *CostGrow* and *Schumann14* have better performance than the simple algorithms and the coarse hydrodynamic model; however, the performance of *CostGrow* slightly surpasses that of *Schumann14*. Given the more comparable inundation performance, we conclude the advantage seen here emerges from *CostGrow's* application of bilinear resampling as opposed to *Schumann14's* nearest-neighbour resampling; however, owing to the relatively small scale ratios (4:32), this advantage is minor. Comparing the simple algorithms in Fig. 7 (panel c0 and a1) shows that the treatment of wet-partial regions provides no improvement in reproducing high water marks, unlike the advantages seen for inundation performance. We hypothesize this owes to the absence of any high water mark observations on dry cells in wet-partial regions. In other words, *TerrainFilter* only improves the filtering of False Positives when compared to the *Resample* result and False Positive regions can not be evaluated by high water mark observations (as these regions are dry in reality).

Focusing on a small region, Fig. 8 shows a portion of the domain where floodwaters likely flowed behind a highway embankment along a small frontage road travelling underneath an overpass (point E). Because the observed inundation layer was mapped primarily by air (Landesamt für Umwelt Rheinland-Pfalz, 2022), this observation data shows the area underneath the overpass as dry; therefore any simulated inundation in this area is marked False Positive (FP) — supporting our hypothesis that the observed inundation is slightly conservative. Other performance and behavioural differences between the algorithms can also be seen in this area: the lack of treatment for wet-partials (point A) and dry-partials (point B) in the simple algorithms; how *CostGrow* is not limited to a search radius like *Schumann14* for dry-partial treatment (point C); the isolated inundation artifacts in *Schumann14* (point D); and the blocky result of *Schumann14's* nearest-neighbour resampling in wet-wet (WW) and wet-partial (WP) regions (panel b1).

Runtimes for the twin hydrodynamic models and the four downscaling algorithms are shown in Table 2. As expected, the algorithms of higher complexity also have higher runtimes and all downscaling algorithms are substantially faster than the hydrodynamic models (which simulate a flood wave of 833 mins). Despite minimal effort being invested in optimization, the novel *CostGrow* algorithm is substantially faster than the state-of-the-art algorithm from Schumann et al. (2014). These runtimes may be improved through parallelization and other programming improvements. Regardless, because Schumann et al. (2014)'s algorithm employs a cell-by-cell nearest-neighbour search, this algorithm is fundamentally less efficient than those like *CostGrow* which employ least-cost mapping.

Comparing alternate approaches to obtain fine resolution flood grids (4 $m$ in this case), the total runtime of a pipeline implementing the *CostGrow* downscaling algorithm (on top of the coarse hydrodynamic model) was roughly 33 seconds versus the 34 minutes necessary for our 4 $m$ native hydrodynamic model, a 60-fold improvement. This reduced runtime has a corresponding loss in temporal resolution (only the maximum WSE is downscaled) and inundation accuracy of 0.03 CSI and high water mark accuracy of 0.14 RMSE when comparing the *CostGrow* downscaling algorithm pipeline to the 4 $m$ native hydrodynamic model for our study. Were a slower hydrodynamic model used (e.g., a non GPU-parallelized platform) or a larger downscaling ratio (32:4 in our case) the efficiency gain of downscaling over fine hydrodynamic modelling would increase; while shortening the simulation time (833 mins in our case) would reduce the efficiency gain. For example, during study development we implemented similar twin hydrodynamic models in the LISFLOOD-FP 8.1 framework using a second-order discontinuous Galerkin solver which implements the full shallow water equations (Shaw et al., 2021). Executed on 8 CPU cores the runtime for these models was 7100.00 mins and 0.43 mins for the 4 $m$ and 32 $m$ discretizations respectively — were a *CostGrow* downscaling algorithm pipeline implemented with this setup we estimate a 16,000-fold improvement in runtime to obtain a comparable $WSE_{s1=4m}$ grid.

The flood grid downscaling algorithms presented here make estimates based on simple hydraulic assumptions and the $DEM$. Because of this, these algorithms do not consider sub-grid or other hydraulically relevant elements not contained in the DEM like levees, flood-walls, or storm drainage systems. The significance of this limitation will depend on the particular case, but any study where levees or barriers are present but *not* resolved by the $DEM$ should be extra cautious when employing downscaling. Much like hydrodynamic models, sub-grid obstructions like levees could be incorporated into downscaling algorithms, e.g., using a non-neutral cost surface in *CostGrow's* cost-distance routine.

Future work should explore evaluation techniques for flood-related algorithms, where the techniques are less sensitive to study area, domain size, hardware, and software of a particular study. For example, a collection of fully-open data-rich flood events with a wide range of hydraulic character would facilitate more meaningful comparisons of model or algorithm performance across platforms and between researchers. Further, such additional case studies with varied hydraulic character would help to quantify and communicate the benefits and limitations of downscaling in different hydraulic regimes. To better support the emerging needs of impact forecasting, where 2D-velocity grids are sometimes desired, pursuing a method that is also capable of downscaling velocity could be of use; however, this would require blocking-out buildings in the hydrodynamic model. Further performance enhancements of downscaling methods may be found by incorporating machine learning tech-

niques originally developed for image enhancement which have recently been applied to enhance terrain models (Demiray et al., 2021).

## 5 Conclusions

This study has developed, demonstrated, and evaluated the novel *CostGrow* algorithm for resolution enhancement or downscaling of flood water surface grids. This algorithm outperforms the state-of-the-art, with a six-fold improvement in runtime for our case study, a slight improvement in standard performance metrics, and improvements in some fringe areas using qualitative evaluation. When compared to results obtained through fine resolution hydrodynamic modelling, the proposed downscaling algorithm (in conjunction with coarse resolution modelling) showed a 60-fold improvement in runtimes with a slight loss of accuracy.

In general, coarse modelling in conjunction with downscaling is shown to be an effective means of obtaining fine resolution inundation grids at a fraction of the computational cost. However, the utility of employing downscaling to obtain fine resolution grids is limited by the availability and quality of fine resolution DEMs. A potential application of downscaling is to facilitate the post-processing of fine resolution inundation results layers from global models for data-rich regions where a local fine resolution DEM is available. This could be a cost effective way to deliver fine resolution inundation maps to any region on the globe without the need for specialized modelling expertise or resources. Regardless, the lack of attention to the subject of downscaling in the academic literature suggests some space remains for downscaling to improve the efficiency of model chains. Towards this, the downscaling algorithms developed in this study have been made open source and are available as QGIS processing scripts (https://github.com/cefect/FloodRescaler).

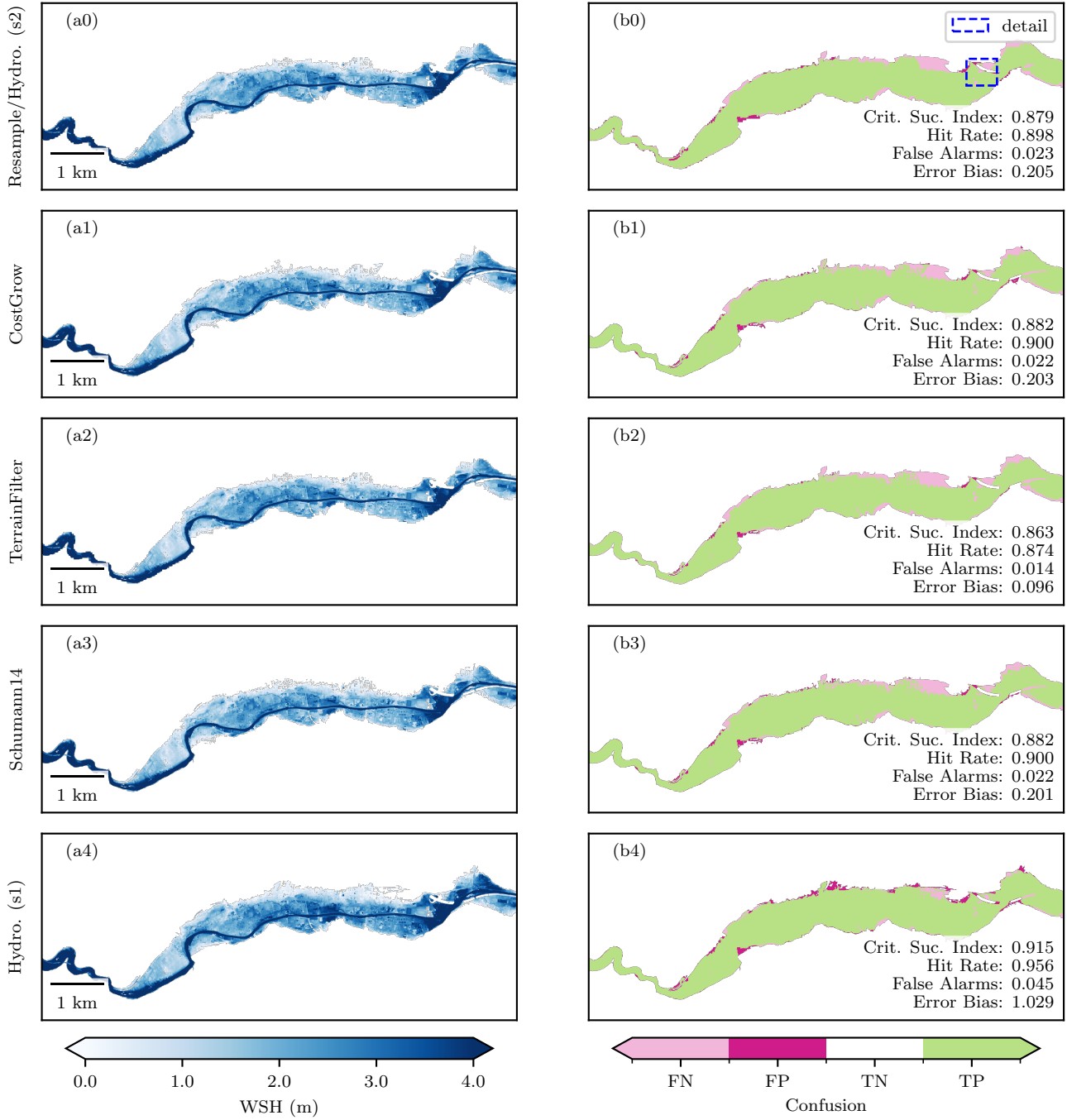

**Figure 6.** Downscaled and hydrodynamic model $WSH_{s1}$ results and corresponding inundation performance. Left-side panels (a) show the water depth grids ($WSH_{s1}$) obtained from the coarse and fine hydrodynamic model (a0 and a4) and the four downscaling algorithms (a0-a3; see Table 2 and text for description). Similarly, the right-side panels (b) show the domain inundation classification confusion map (False Negative (FN), False Positive (FP), True Negative (TN), and True Positive (TP) cells – see Table S1) and the common inundation performance metrics defined in Table S2. The *Resample* algorithm and coarse hydrodynamic model results ($s2$) are identical and therefore shown as one panel. See Fig. 8 for detail area.

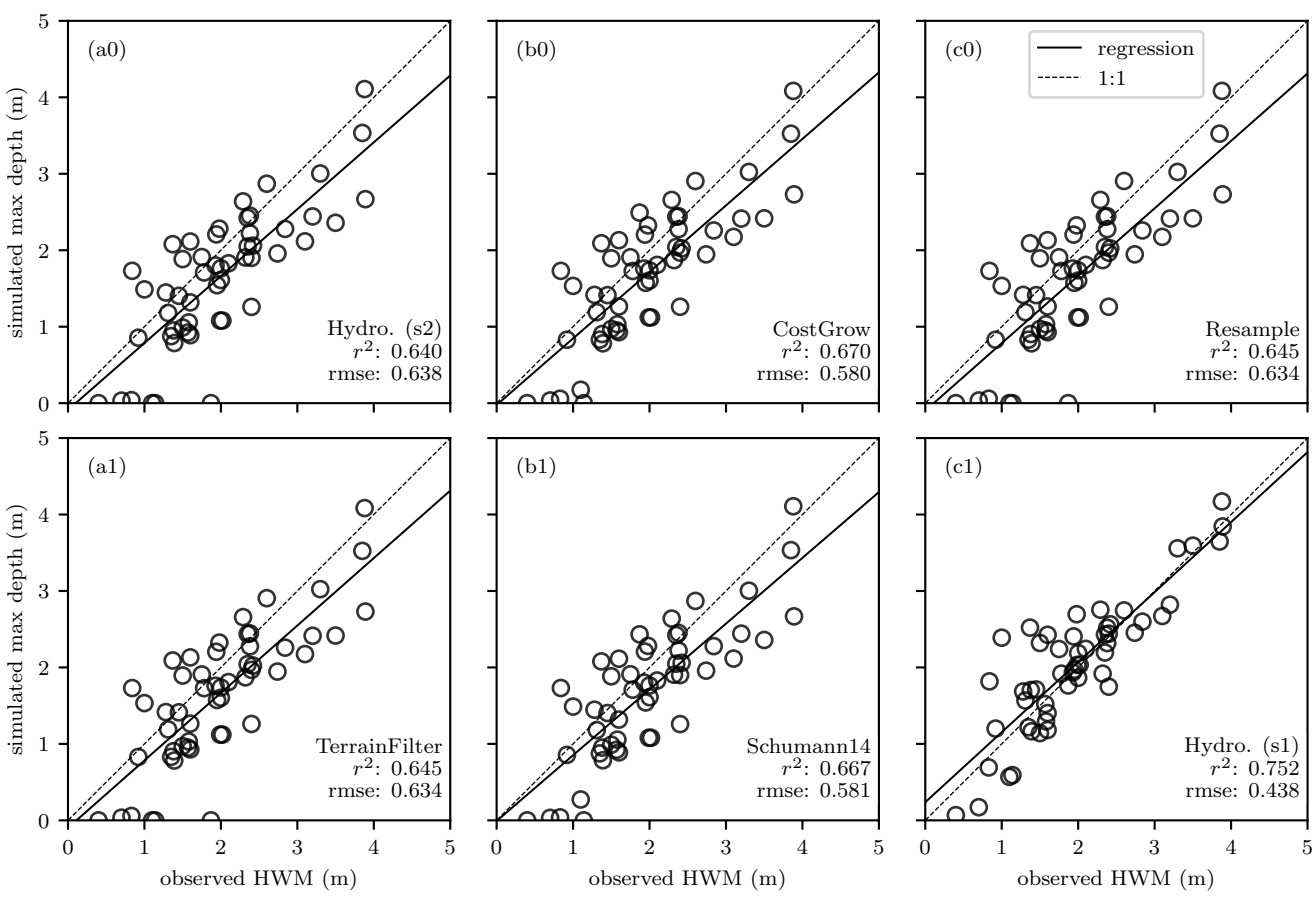

**Figure 7.** Linear correlation between July 2021 resident-reported high water marks (see Table 1) and maximum simulated depth of twin hydrodynamic models (a0, c1) and downscaling algorithms (b0, c0, a1, b1).

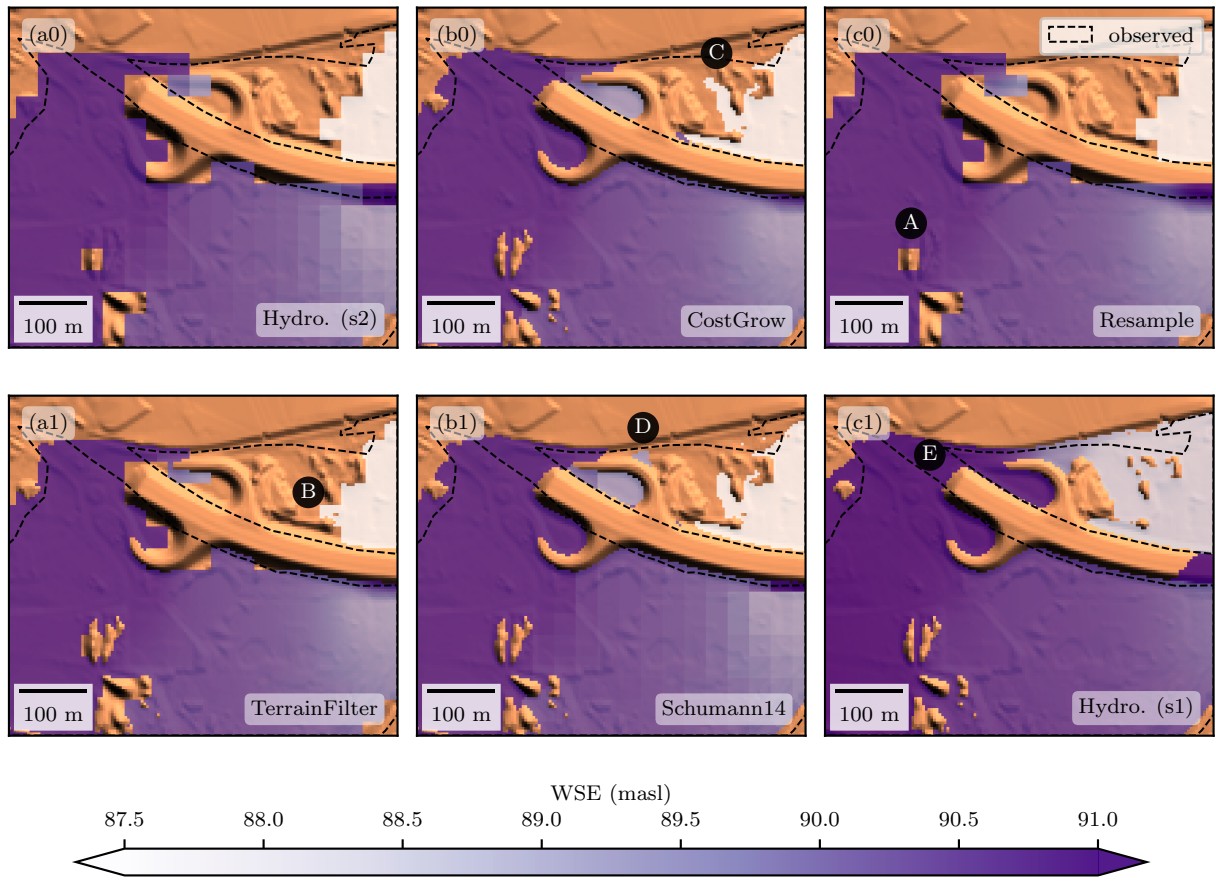

**Figure 8.** Downscale and hydrodynamic model $WSE$ detail results. Observed inundation is shown in black for reference. See Fig. 6b0 for location.

*Code availability.* Scripts used in downscaling analysis and figure production are provided by Bryant (2023). Easy-to-use QGIS Processing scripts (not used for this publication) are provided in the FloodRescaler project (https://github.com/cefect/FloodRescaler).

320 *Data availability.* Case study data used to construct the hydrodynamic models and perform the validation are summarized in Table 1 and the corresponding references (all data except the high water marks are public). Grids used in the downscaling analysis are available upon request.

*Author contributions.* SB developed the concept, analysis, and writing with contributions from other co-authors. SB developed the software, formal analysis, and investigation. GS provided the MATLAB source code for the *Schumann14* algorithm. HA requested and managed the
325 calibration data and provided early versions of the hydrodynamic models. BM and HK supervised the work.

*Competing interests.* The authors declare that they have no conflicts of interest.

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
