# Peer review of "Technical Note: Resolution Enhancement of Flood Inundation Grids"

_Hydrology and Earth System Sciences, 2023_

## Author Comment (AC1)

**Technical Note: Resolution Enhancement of Flood Inundation Grids**

Response to Reviewer #1

**Seth Bryant, Guy Schumann, Heiko Apel, Heidi Kreibich, Bruno Merz**

**Friday, October 20, 2023**

We thank the editor and the reviewers for their comments. These have been very helpful in improving the manuscript. Responses to the major comments are provided below (along with the original comment in italics). Some responses to minor comments are also provided where necessary (other minor comments will simply be incorporated into the manuscript).

**Reviewer #1**

> *In my view, the strongest result, and one that best showcases the superiority of the new algorithm, is in Figure 6. It shows that the new algorithm can much better represent small-scale inundation extent. This is very important and, clearly, cannot be well captured by standard evaluation metrics. I encourage the author to emphasize this result (including in the abstract and conclusions) and provide additional examples (using the same flood case study). Do not shy away from qualitative comparison.*

This is a nice point which strengthens the manuscript that we had not considered. To incorporate this comment, the following sentence will be added to the abstract:

> Qualitatively, the algorithm generates more physically coherent flood maps in some hydraulically challenging regions compared to the state-of-the-art.

Similarly, the following sentence will be added to the Results and Discussion section along with the accompanying figure in the supplement:

> Qualitatively however, *CostGrow* generates more physically coherent depth grids in some fringe areas as shown in Figure 8 and S8.

And finally, the following sentence in the conclusions will be modified:

> This algorithm outperforms the state-of-the-art, with a six-fold improvement in runtime for our case study, a slight improvement in standard performance metrics, and improvements in some fringe areas using qualitative evaluation.

*Adding building impact may also help showcase the improved capability at small scales.*

We considered this and agree that this may also better demonstrate the improvement provided by the new algorithm; however, we felt the added complexity and length required for this comparison were not justified in a Technical Note.

*The new algorithm's description needs to be extended.*

This section will be expanded and the sub-routines elaborated as follows:

The novel CostGrow algorithm employs the four phases summarized in Fig. 1: 1) grid resampling; 2) least-cost mapping; 3) filtering high-and-dry sub-grid cells; and finally 4) an isolated-cell filter all of which are parameterless. In the first grid resampling phase, various techniques have been developed by others for applications in image analysis and spatial analysis (Bierkens et al., 2000) with bilinear being the most common for terrain manipulations in hydraulic applications (Heritage et al., 2009; Muthusamy et al., 2021) as it provides a smooth result while preserving centroid values. For downscaling, bilinear resampling computes the s1 value from the four adjacent s2 centroid values weighted by distance as seen in Fig. 1a (notice the s2 values are preserved by the center s1 cell). CostGrow implements bilinear resampling from the popular spatial analysis package GDAL (GDAL/OGR contributors, 2022). In the second phase, the resampled grid is extrapolated using a cost-distance analysis, a common GIS algorithm for computing the path of least cost, determined by weighting distance and some cost map to obtain the effective distance from source cells to sink cells Foltête et al. (2008). For this study CostGrow implements a cost-distance routine with N8(P) adjacency and a neutral cost surface (Lindsay, 2014, CostAllocation). This first maps the dry portion of the domain in terms of catchment areas for each boundary W SEs1,i from the previous phase, then maps corresponding boundary W SEs1,i value to each of its catchments. In effect, this grows each W SEs1,i boundary cell value outwards, filling the dry domain with the W SEs1,i values that are closest in distance. For the toy example shown in Fig. 1b, this is a simple extrapolation onto the dry right-side of the domain. Future implementations could employ a non-neutral cost surface to incorporate levees or some other flood obstructions into the analysis. In the third phase, high-and-dry cells are filtered from this cost-distance map by comparing cell-by-cell to the terrain values (where DEMs1 > W SEs1,i set W SEs1,i = NULL) as shown by the blank cells in Fig. 1c. This often results in many isolated pockets of flooding in low-lying areas shown beyond the initial contiguous W SEs2 flood which are addressed below (see Fig. 1c red circle). In the final phase, isolated or disconnected groups of flooded cells are filtered from the result such that only the largest or main flooded water body remains. To accomplish this, the filtered grid is converted to a binary inundation grid, from which each contiguous clump is identified and ranked according to size (Lindsay, 2014, Clump) (see Fig. 1d). From the largest clump, an inverted mask is generated and applied to the water level grid to remove isolated flooding cells from the result.

*Line 177 – what was the performance of s2?*

This section discusses the inundation performance metrics. Line 174 states '…the results for the Resample downscaling algorithm (Fig. 6a0 and b0) show the performance of the coarse (s2) parameterization'. The original figure is pasted below.

[Figure]

[Figure]

*Figure 5 – 'rvalue' – is this R or R2? R2 should be reported. Explain the observations with values close to 0.*

The figure will be revised to show $r^2$. The simulations close to zero are for HWM locations where the simulated extents predict dry (False Negative). This explanation will also be provided in the manuscript.

---

## Author Comment (AC2)

**Technical Note: Resolution Enhancement of Flood Inundation Grids**

Response to Reviewer #2

**Seth Bryant, Guy Schumann, Heiko Apel, Heidi Kreibich, Bruno Merz**

**Friday, October 20, 2023**

We thank the editor and the reviewers for their comments. These have been very helpful in improving the manuscript. Responses to the major comments are provided below (along with the original comment in italics). Some responses to minor comments are also provided where necessary (other minor comments will simply be incorporated into the manuscript).

**Reviewer #2**

*My first comment is that this manuscript depends too much on supplemental material. Figure S1 presents the synthetic case and a table used to define the possible cases one may encounter when downscaling, that is the graphical support to a conceptualization. Figure S2 and Table S1 present the case-study. Even though the supplemental material is available, I think that crucial information needs to be in the main manuscript. I do not know why the authors opted for this way, probably to fit some limitation length, but the result is unsatisfactory because no way the manuscript can be a stand-alone product.*

We feel that the manuscript is able to stand alone without these — at some loss of readability. For example, Figure S1 (the conceptual framework) is a graphical representation of Equation (1): having the graphical version obviously makes the concept easier to grasp but is not strictly necessary to communicate the framework. Similarly, the case-study support Figure S2 and Table S1 are helpful to explain and better define the case study; however, they are not strictly necessary for the reader to understand that we used a case study to demonstrate the relative performance of the algorithm. Regardless, following the advice of the reviewer, we will move these figures into the main body of the manuscript to improve readability.
* * *
*Second, I think that a conceptual discussion to introduce the challenges related to the work needs to be added. Presently, it is left to figure S1 and eq. (1) that introduce the easy and the "partial" cases. I think that more emphasis should be given to the removal of cells that would be wet after the simple interpolation (the "isolated cell*

*filter" is not well described) and to the extrapolation of inundation (that is another crucial step to correct the bare result of the interpolation).*

The following will be added to the Resample Case Framework section to better present the problem of downscaling and discuss isolated flooding and the inundation extrapolation:

Beginning with the simplest case, dry-dry (DD) zones are trivial for rescaling operations as they remain dry before and after rescaling and can therefore remain unaltered. Wet-wet (WW) zones are also relatively simple as the group of WSE_i cells should be roughly equivalent to their parent WSE_j cell; an easy task for classic grid resampling tools like bilinear resampling. Wet-partial (WP) zones can be similarly obtained, but with the additional step of removing dry s1 cells with an exceedance mask (i.e., cells where WSE_s1<DEM_s1 are removed). More difficult is the treatment of dry-partial (DP) zones which require some propagation or searching beyond the original coarse (s2) inundation footprint. In fact, this propagation challenge is a similar problem to that of a classic 2D hydrodynamic model with a small domain; however, downscaling must employ simplifying assumptions to maintain advantageous computation times. This requires more sophisticated algorithms that may introduce artifacts like isolated flooding where disconnected groups of cells in low-lying terrain are shown as flooded (e.g., behind levees). Because high-resolution hydrodynamic models generally employ cell-connected routines, this isolated flooding can be considered erroneous within the paradigm of hydrodynamic downscaling. However, this paradigm can be a poor representation of actual flood behaviour in areas with high groundwater connectivity or permeable flood defences. In summary, the downscaling problem can be broken into three zones: the first two (WW and WP) are relatively simple and, while some alternate approaches are possible, we do not expect large differences in performance in these zones. The final zone is more challenging and we therefore expect differences in performance between algorithms in this dry-partial (DP) zone.
* * *
*Third, the methods are not well described. In the description of the CostGrow, what is a cost and how is it minimized? How is an isolated clump recognized? And so on. In addition, figures do not help. For example, values in the panels of fig 1 are not described, not even defined. I guess that panels (abc) contain elevations while panel (d) contains codes that will mean something (clump name?). Furthermore, overlapping wse and DEM in panel a does not help. It would be better to have panels up to (e) to see everything (I know that the DEM is also in fig 2a but we need it here). Also it would be helpful to make examples of what happens in some cells. Commenting some output would be important for understanding how the method works.*

The *Novel CostGrow Algorithm* section will be expanded to better explain the cost-distance analysis and isolated cell filter routines. Annotation of the toy results has also been included in this section. The revised paragraph is below:

The novel CostGrow algorithm employs the four phases summarized in Fig. 1: 1) grid resampling; 2) least-cost mapping; 3) filtering high-and-dry sub-grid cells; and finally 4) an isolated-cell filter all of which are parameterless. In the first grid resampling phase, various techniques have been

developed by others for applications in image analysis and spatial analysis (Bierkens et al., 2000) with bilinear being the most common for terrain manipulations in hydraulic applications (Heritage et al., 2009; Muthusamy et al., 2021) as it provides a smooth result while preserving centroid values. For downscaling, bilinear resampling computes the s1 value from the four adjacent s2 centroid values weighted by distance as seen in Fig. 1a (notice the s2 values are preserved by the center s1 cell). CostGrow implements bilinear resampling from the popular spatial analysis package GDAL (GDAL/OGR contributors, 2022). In the second phase, the resampled grid is extrapolated using a cost-distance analysis, a common GIS algorithm for computing the path of least cost, determined by weighting distance and some cost map to obtain the effective distance from source cells to sink cells Foltête et al. (2008). For this study CostGrow implements a cost-distance routine with N8(P) adjacency and a neutral cost surface (Lindsay, 2014, CostAllocation). This first maps the dry portion of the domain in terms of catchment areas for each boundary $WSEs1,i$ from the previous phase, then maps corresponding boundary $W SEs1,i$ value to each of its catchments. In effect, this grows each $W SEs1,i$ boundary cell value outwards, filling the dry domain with the $W SEs1,i$ values that are closest in distance. For the toy example shown in Fig. 1b, this is a simple extrapolation onto the dry right-side of the domain. Future implementations could employ a non-neutral cost surface to incorporate levees or some other flood obstructions into the analysis. In the third phase, high-and-dry cells are filtered from this cost-distance map by comparing cell-by-cell to the terrain values (where $DEMs1 > W SEs1,i$ set $W SEs1,i = NULL$) as shown by the blank cells in Fig. 1c. This often results in many isolated pockets of flooding in low-lying areas shown beyond the initial contiguous $W SEs2$ flood which are addressed below (see Fig. 1c red circle). In the final phase, isolated or disconnected groups of flooded cells are filtered from the result such that only the largest or main flooded water body remains. To accomplish this, the filtered grid is converted to a binary inundation grid, from which each contiguous clump is identified and ranked according to size (Lindsay, 2014, Clump) (see Fig. 1d). From the largest clump, an inverted mask is generated and applied to the water level grid to remove isolated flooding cells from the result.

As for the figures, you have understood them correctly. To further clarify, we will duplicate the input grids to Figure 1 as shown below.

[Figure]

The figure captions will also be re-written to clarify that the values seen are grid values.

*Fourth, I think that it would be important to stress that the method will refine and correct the inundation map up to what can be seen by the refined DEM. In many cases, particularly in urban contexts, the pattern of the inundation is determined by narrow objects (typically, narrow bank walls) that are invisible even in DEMs with a resolution of 1 m. A correct representation of the effects of these structures on the inundation dynamics requires a specific treatment in the hydrodynamic model. In cases where thin structures matter, I would not recommend using a downscaling algorithm. I trust that this could be irrelevant for the case-study presented here, but the authors may add their view on this or similar issues.*

This is an important point we agree with. The following paragraph will be added to the Analysis section to highlight this limitation for the reader:

> The flood grid downscaling algorithms presented here make estimates based on simple hydraulic assumptions and the DEM. Because of this, these algorithms do not consider sub-grid or other hydraulically relevant elements not contained in the DEM like levees, flood-walls, or storm drainage systems. The significance of this limitation will depend on the particular case, but any study where levees are present but not resolved by the DEM should be extra cautious when employing downscaling. Much like hydrodynamic models, sub-grid obstructions like levees

could be incorporated into downscaling algorithms (because of CostGrow's cost-distance routine this feature should be easy to implement).
* * *
*Fifth, I am somehow puzzled by figure S5, where the concentration of red at the right and blue at the left of the picture would mean that changing the resolution of the hydrodynamic model changes significantly the mean slope of the free surface along the reach, in turn indicating that at least one of the models will not represent correctly the reality.*

The figure shows differences in elevation between the coarse (s2) and fine (s1) calibrated hydrodynamic models. As you mention, the fine water levels downstream (panel right) were 0 to 1m higher than the coarse model. The following sentences will be added to the supplement to discuss this trade-off:

> While this difference would be problematic for some hydrodynamic model applications, here we focus on inundation extents – not elevations. Because of this, we use a simple calibration of two roughness parameters to optimize the Critical Success Index which is a measure of fit to the observed inundation extents. It is therefore not surprising that inundation is reproduced well by both models while water elevations are less satisfactory. A more sophisticated (e.g., multi-metric optimization) calibration could have been pursued to try and address this; however, as our paper focuses on downscaling (not model calibration) we felt this would be distracting.

*74: This paper in under review. More information is needed here to understand the (synthetic?) case. This further supports a request to move material from the supplemental to the manuscript.*

The References section will be updated to reflect that this paper has since been published.
* * *
*86: May be not just a matter of using an exceedance mask. in case the water is confined by an embankment and the coarse-mesh model returns a wide inundated area, the exceedance mask will dry the cells corresponding to the embankment but not the external ones. In this case, it is a matter of drying cells disconnected from the inundation area after the application of the exceedance mask. To me this is related to the second major comment and a need to give more emphasis to the removal of isolated flooded regions that will come with figure 1.*

This function is provided by 'isolated filter' step of CostGrow. This is further clarified by the above response to comment #3 (which proposes to expand the description of the algorithm).
* * *
*98-109: already mentioned above that the description of the method would need to be richer.*

See response to comment #3.
* * *
*114: I am really shocked reading that the case-study flood had a return period ranging from 3 thousand to 60 thousand years. Without an explanation this statement just sounds crazy. I would suggest to provide some details to mention where these huge and very different values come from, or to remove the statement that is not crucial for the assessment of the algorithm's performance.*

This statement will be removed.

*118: "calibrated to the oberved inundation" is unclear. Which is the calibration parameter, and which the criterion? Later it will be mentioned that the calibration is aimed at reproducing the extent of the inundation (not the depths) by changing two roughness coefficients (river and urban), but the information is needed here.*

As you mention, a full paragraph on the next page is provided to explain the calibration routine (along with a section in the supplement). The sentence referred to here is a summary and therefore does not include all such details. Regardless, it will be expanded to read as follows:

To construct the coarse water grid for use as an input in downscaling (WSE_s2) and a second grid for validation and comparison (WSE_s1), coarse (s2=32m) and fine (s1=4m) resolution twin hydrodynamic models are calibrated to the observed inundation using the Critical Success Index.

*Figure 1: why a subscript B? Before they were i and j.*

$WSE_{s1,B}$ refers to the bilinear full-domain **grid** data layer**,** while $WSE_{s1,i}$ refers to an individual cell within a grid.

*125: this sounds strange. Avoiding a presumably better treatment because it is affected by resolution is kind of weird in a work aimed at resolution enhancement and providing (line 1) "high resolution flood maps". Sounds like this intends to be a preliminary study and validation in "easy" cases.*

This section was poorly worded and has been revised as shown below.

For the treatment of urban areas, blocking-out buildings has been shown to be a more accurate way to represent buildings within a relatively high-resolution hydrodynamic model (Bellos 2015). In coarse models however, especially where the building size (and space between) is smaller than a grid-cell, blocking-out reduces model performance, especially in and around buildings. Regardless, blocking-out could be included into a downscaling algorithm (and the high-resolution validation model); however, as this requires a more complicated algorithm and is not included in the current state-of-the-art methods against which we compare, we opted to avoid blocking-out and instead apply a separate roughness coefficient to built-up areas to capture the blocking effects of buildings.

*131: here and elsewhere, the measuring units for the Manning coefficient should be added.*

Units of $s/m^{1/3}$ will be added to the text.

*137: 0.867 > 0.175, ok; 0.089 < 0.133, strange (considering how the n should change with grid size). Can a comment be added in this respect?*

We understand you are referring to the counterintuitive result of the calibration for the roughness of the 32m (0.867 and 0.089) vs. the 4m (0.175 and 0.133) models. The following sentence will be added to address this:

> These counterintuitive relative roughnesses are likely a result of differences in floodplain-channel dynamics necessary to match the observed inundation footprint between the two models.

*141: Here a discussion of the calibration was presented only for the hydrodynamic model. What about cost-grow, is it parameterless?*

Yes, the algorithm is parameterless. This will be clarified in the algorithm description.

*164: this statement may induce in a reader a doubt that the result presented here comes from the authors' interpretation of a previous model, and may thus suffer from misinterpretation. In my understanding, this should not be the case since a Schumann is also in the authorship of the present paper, which may be made explicit.*

The section has been clarified to indicate that our interpretation comes from the source code.

*207: here we find again a detail on model calibration that should have come much earlier.*

Both points mentioned here in the discussion are first presented in the methods (optimization target) or Table S1 (HWMs).

---

## Author Response (AR1)

**Technical Note: Resolution Enhancement of Flood Inundation Grids**

Seth Bryant, Guy Schumann, Heiko Apel, Heidi Kreibich, Bruno Merz

Tuesday, November 28, 2023

We thank the editor and the reviewers for their comments. These have been very helpful in improving the clarity of the manuscript. Responses to the major comments are provided below (along with the original comment in italics). Some responses to minor comments are also provided where warranted, while other minor comments were simply incorporated into the manuscript.

This letter is similar to the one submitted on Oct. 20th but has been revised slightly to reflect the latest changes.

**Reviewer #1**

> *In my view, the strongest result, and one that best showcases the superiority of the new algorithm, is in Figure 6.  It shows that the new algorithm can much better represent small-scale inundation extent. This is very important and, clearly, cannot be well captured by standard evaluation metrics. I encourage the author to emphasize this result (including in the abstract and conclusions) and provide additional examples (using the same flood case study). Do not shy away from qualitative comparison.*

This is a nice point which strengthens the manuscript that we had not considered. To incorporate this comment, the following sentence will be added to the abstract:

> Qualitatively, the algorithm generates more physically coherent flood maps in some hydraulically challenging regions compared to the state-of-the-art.

Similarly, the following sentence will be added to the Results and Discussion section along with the accompanying figure in the supplement:

> Qualitatively however, *CostGrow* generates more physically coherent depth grids in some fringe areas as shown in Figure 8 and S8.

And finally, the following sentence in the conclusions will be modified:

> This algorithm outperforms the state-of-the-art, with a six-fold improvement in runtime for our case study, a slight improvement in standard performance metrics, and improvements in some fringe areas using qualitative evaluation.

*Adding building impact may also help showcase the improved capability at small scales.*

We considered this and agree that this may also better demonstrate the improvement provided by the new algorithm; however, we felt the added complexity and length required for this comparison were not justified in a Technical Note.

*The new algorithm's description needs to be extended.*

This section has been substantially extended.

*Line 177 – what was the performance of s2?*

This section discusses the inundation performance metrics. The performance of s2 is stated on Line 174 '[…] the results for the Resample downscaling algorithm (Fig. 6a0 and b0) show the performance of the coarse (s2) parameterization'. The original figure is pasted below.

[Figure]

[Figure]

*Figure 5 – 'rvalue' – is this R or R2? R2 should be reported. Explain the observations with values close to 0.*

The figure has been revised to show $r^2$. The simulations close to zero are for HWM locations where the simulated extents predict dry (False Negative). This explanation is now provided in the manuscript.

**Reviewer #2**

*My first comment is that this manuscript depends too much on supplemental material. Figure S1 presents the synthetic case and a table used to define the possible cases one may encounter when downscaling, that is the graphical support to a conceptualization. Figure S2 and Table S1 present the case-study. Even though the supplemental material is available, I think that crucial information needs to be in the main manuscript. I do not know why the authors opted for this way, probably to fit some limitation length, but the result is unsatisfactory because no way the manuscript can be a stand-alone product.*

Following the advice of the reviewer, these figures have been moved into the main body of the manuscript to improve readability.
* * *
*Second, I think that a conceptual discussion to introduce the challenges related to the work needs to be added. Presently, it is left to figure S1 and eq. (1) that introduce the easy and the "partial" cases. I think that more emphasis should be given to the removal of cells that would be wet after the simple interpolation (the "isolated cell filter" is not well described) and to the extrapolation of inundation (that is another crucial step to correct the bare result of the interpolation).*

The Resample Case Framework section has been extended and revised to better discuss these challenges/steps.
* * *
*Third, the methods are not well described. In the description of the CostGrow, what is a cost and how is it minimized? How is an isolated clump recognized? And so on. In addition, figures do not help. For example, values in the panels of fig 1 are not described, not even defined. I guess that panels (abc) contain elevations while panel (d) contains codes that will mean something (clump name?). Furthermore, overlapping wse and DEM in panel a does not help. It would be better to have panels up to (e) to see everything (I know that the DEM is also in fig 2a but we need it here). Also it would be helpful to make examples of what happens in some cells. Commenting some output would be important for understanding how the method works.*

The section Novel CostGrow Algorithm has been expanded to better explain the cost-distance analysis and isolated cell filter routines. Annotation of the toy results have also been included in this section.

As for the figures, you have understood them correctly. To further clarify, we have duplicated the input grids to Figure 1. The figure caption has also been re-written to clarify that the values seen are grid values.
* * *
*Fourth, I think that it would be important to stress that the method will refine and correct the inundation map up to what can be seen by the refined DEM. In many cases, particularly in urban contexts, the pattern of the inundation is determined by narrow objects (typically, narrow bank walls) that are invisible even in DEMs with a resolution of 1 m. A correct representation of the effects of these structures on the inundation dynamics requires a specific treatment in the hydrodynamic model. In cases where thin structures matter, I would not recommend using a downscaling algorithm. I trust that this could be irrelevant for the case-study presented here, but the authors may add their view on this or similar issues.*

This is an important point we agree with. A paragraph has been added to the Analysis section to highlight this limitation for the reader.
* * *
*Fifth, I am somehow puzzled by figure S5, where the concentration of red at the right and blue at the left of the picture would mean that changing the resolution of the hydrodynamic model changes significantly the mean slope of the free surface along the reach, in turn indicating that at least one of the models will not represent correctly the reality.*

The figure shows differences in elevation between the coarse (s2) and fine (s1) calibrated hydrodynamic models. As you mention, the fine water levels downstream (panel right) were 0 to 1m higher than the coarse model. The following sentences have been added to the supplement to discuss this trade-off:

> While this difference would be problematic for some hydrodynamic model applications, here we focus on inundation extents – not elevations. If elevations and flow dynamics were the focus, the coarse hydrodynamic model would be an inappropriate choice, because the model resolution is about 3 times larger than the width of the river, which results in the observed deviations in water slope profile compared to the fine resolution model. For an estimation of the flood extent, however, the coarse model can provide useful results despite the deficiencies in simulating the flow dynamics. Because of our focus on flood extent, we use a simple calibration of two roughness parameters to optimize the Critical Success Index which is a measure of fit to the observed inundation extents. It is therefore not surprising that inundation is reproduced well by both models while water elevations are less satisfactory. A more sophisticated (e.g., multi-metric optimization) calibration could have been pursued to try and address this; however, as our paper focuses on downscaling (not model calibration) we felt this would be distracting.

*74: This paper in under review. More information is needed here to understand the (synthetic?) case. This further supports a request to move material from the supplemental to the manuscript.*

The References section has been updated to reflect that this paper has since been published.

*86: May be not just a matter of using an exceedance mask. in case the water is confined by an embankment and the coarse-mesh model returns a wide inundated area, the exceedance mask will dry the cells corresponding to the embankment but not the external ones. In this case, it is a matter of drying cells disconnected from the inundation area after the application of the exceedance mask. To me this is related to the second major comment and a need to give more emphasis to the removal of isolated flooded regions that will come with figure 1.*

This function is provided by 'isolated filter' step of CostGrow. This point has been further clarified by our response to Reviewer #1's third comment (which expands the description of the algorithm).

*98-109: already mentioned above that the description of the method would need to be richer.*

This section has been expanded.
* * *
*114: I am really shocked reading that the case-study flood had a return period ranging from 3 thousand to 60 thousand years. Without an explanation this statement just sounds crazy. I would suggest to provide some details to mention where these huge and very different values come from, or to remove the statement that is not crucial for the assessment of the algorithm's performance.*

This statement has been removed, because of the problems of estimating the probability of the event by the available gauge time series.

*118: "calibrated to the oberved inundation" is unclear. Which is the calibration parameter, and which the criterion? Later it will be mentioned that the calibration is aimed at reproducing the extent of the inundation (not the depths) by changing two roughness coefficients (river and urban), but the information is needed here.*

As you mention, a full paragraph on the next page is provided to explain the calibration routine (along with a section in the supplement). The has been expanded to clarify the calibration parameter.

*Figure 1: why a subscript B? Before they were i and j.*

$WSE_{s1,B}$ refers to the bilinear full-domain **grid** data layer**,** while $WSE_{s1,i}$ refers to an individual cell within a grid.

*125: this sounds strange. Avoiding a presumably better treatment because it is affected by resolution is kind of weird in a work aimed at resolution enhancement and providing (line 1) "high resolution flood maps". Sounds like this intends to be a preliminary study and validation in "easy" cases.*

This section was poorly worded and has been revised.
* * *
*131: here and elsewhere, the measuring units for the Manning coefficient should be added.*

Units of $s/m^{1/3}$ have been added to the text.

*137: 0.867 > 0.175, ok; 0.089 < 0.133, strange (considering how the n should change with grid size). Can a comment be added in this respect?*

We understand you are referring to the counterintuitive result of the calibration for the roughness of the 32m (0.867 and 0.089) vs. the 4m (0.175 and 0.133) models. The following sentence has been added to address this:

These counterintuitive relative roughnesses are a result of differences in floodplain-channel dynamics necessary to match the observed inundation footprint between the two models. In the coarse hydrodynamic model, the river channel is poorly represented by the 32m resolution which is roughly 3 times larger than the channel. Thus, the flow dynamics in the channel, as well as the channel-floodplain interactions, show different dynamics compared to the more realistic fine resolution model. The calibration routine compensates for these differences with the disproportionate roughness values reported above. However, as our focus is on downscaling performance, the less-accurate representation of channel dynamics and water levels (as opposed to inundation extents) provided by the coarse model are inconsequential considering we apply the fine and coarse hydrodynamic model results comparatively in all scenarios.

*141: Here a discussion of the calibration was presented only for the hydrodynamic model. What about cost-grow, is it parameterless?*

Yes, this has been clarified in the algorithm description.

*164: this statement may induce in a reader a doubt that the result presented here comes from the authors' interpretation of a previous model, and may thus suffer from misinterpretation. In my understanding, this should not be the case since a Schumann is also in the authorship of the present paper, which may be made explicit.*

The section has been clarified to indicate that our interpretation comes from the source code.

*207: here we find again a detail on model calibration that should have come much earlier.*

Both points mentioned here in the discussion are first presented in the methods (optimization target) or Table S1 (HWMs).